# Baseline and Early Predictors of Good Patient Candidates for Second-Line after Sorafenib Treatment in Unresectable Hepatocellular Carcinoma

**DOI:** 10.3390/cancers11091256

**Published:** 2019-08-27

**Authors:** Hitomi Takada, Masayuki Kurosaki, Kaoru Tsuchiya, Yasuyuki Komiyama, Jun Itakura, Yuka Takahashi, Hiroyuki Nakanishi, Yutaka Yasui, Nobuharu Tamaki, Chiaki Maeyashiki, Shun Kaneko, Kenta Takaura, Mayu Higuchi, Mao Okada, Wan Wang, Leona Osawa, Shuhei Sekiguchi, Yuka Hayakawa, Koji Yamashita, Nobuyuki Enomoto, Namiki Izumi

**Affiliations:** 1Department of Gastroenterology and Hepatology, Musashino Red Cross Hospital, 1-26-1 Kyonan-cho, Musashino-shi, Tokyo 180-8610, Japan; 2First Department of Internal Medicine, Faculty of Medicine, University of Yamanashi, Yamanashi 409-3898, Japan; 3Department of Gastroenterology and Hepatology, Municipal Hospital of Kofu, Yamanashi 400-0832, Japan

**Keywords:** hepatocellular carcinoma, tyrosine kinase inhibitor, Child–Pugh score, albumin–bilirubin grade

## Abstract

*Background:* Recent advances in the development of tyrosine kinase inhibitors (TKIs) have enabled patients with unresectable hepatocellular carcinoma (HCC) to receive multiple TKIs in sequence. The aim of this study was to identify predictors of good candidates for second-line treatment after disease progression during sorafenib treatment. *Methods:* This is a retrospective cohort study of 190 consecutive HCC patients who were treated with sorafenib in our hospital. Three criteria of good candidates for second-line TKI at the time of disease progression during sorafenib treatment were defined as follows: criterion 1 was the same as the inclusion criteria of the regorafenib for patients with hepatocellular carcinoma who progressed on sorafenib treatment (RESORCE) study, criterion 2 was the inclusion criteria of the RESORCE study plus Child–Pugh score 5, and criterion 3 was the inclusion criteria of the RESORCE study plus albumin–bilirubin (ALBI) grade 1. Factors at baseline and at week 4 during sorafenib treatment were used to predict patients fulfilling each of these three criteria. *Results:* The distribution of patients was 29%, 13%, and 6% in criteria 1, 2, and 3, respectively. Significant factors for meeting criterion 1 was the combination of baseline albumin >3.7 g/dL (odds ratio (OR) 2.7) plus degree of decrease in albumin (Δalbumin) at week 4 <0.2 g/dL (OR 2.6), or the combination of baseline ALBI score <−2.33 (OR 2.5) and ΔALBI at week 4 <0.255 (OR 4.9). For criterion 2, the value of baseline albumin and ALBI score was identical to criterion 1; however, Δalbumin (<0.1 g/dL) and ΔALBI score (<0.19) became stricter. For criterion 3, the value of baseline albumin (>3.8 g/dL) and ALBI (<−2.55) became stricter, as did Δalbumin (<0.1 g/dL) and ΔALBI (<0.085). Furthermore, tumor burden (>11) was selected as an additional predictor (OR 5.4). *Conclusion:* Predictors to satisfy the RESORCE study inclusion criteria were as follows: preserved liver function at baseline, as reflected by albumin or ALBI score, and small deterioration of liver function early during sorafenib therapy, as reflected by Δalbumin or ΔALBI at week 4. Liver function at baseline and degree of change in liver function during sorafenib treatment need to be stricter for better outcomes of liver function with disease progression.

## 1. Introduction

Recent developments in tyrosine kinase inhibitors (TKIs) have increased treatment choices in advanced-stage hepatocellular carcinoma (HCC). For first-line systemic chemotherapy, sorafenib was the first TKI to show a survival benefit over placebo in clinical trials [1,2] and has subsequently been widely used in clinics around the world [3,4,5]. However, treatment choice had been limited in patients who have progressive disease (PD) during sorafenib treatment until the approval of regorafenib as a second-line systemic chemotherapy. The survival benefit of regorafenib over placebo was demonstrated in the phase 3, double-blinded RESORCE study with a hazard ratio of 0.63 [1,2]. In the regorafenib for patients with hepatocellular carcinoma who progressed on sorafenib treatment (RESORCE) study, patients were carefully selected using the following inclusion criteria: (a) oral administration of sorafenib was discontinued due to progressive disease, diagnosed by imaging; (b) hepatic function remained at Child–Pugh A when sorafenib was discontinued due to PD; and (c) patients tolerated 400 mg/day or higher of sorafenib for more than 20 days during the 28 days before the discontinuation of sorafenib. Because the chemical structures of regorafenib and sorafenib are very similar, tolerability to sorafenib is a logical eligibility criterion for regorafenib as a second-line treatment. Conserved liver function within Child–Pugh A is the key inclusion criteria in all clinical trials of systemic therapy for HCC [1,2,6,7,8,9,10] because it is one of the most important determinants of adherence to chemotherapy and survival [3]. Apart from the well-established Child–Pugh classification, recent work suggests that albumin–bilirubin (ALBI) grade [11], a novel classification based on serum albumin and bilirubin, is closely related to liver function reserve and is thus a powerful predictor of prognosis in HCC in various treatment regimens, including sorafenib [12,13,14].

In addition to the already approved regorafenib, second-line treatment options are rapidly expanding. Lenvatinib was demonstrated to be noninferior to sorafenib as first-line chemotherapy in the phase 3 REFLECT trial (Lenvatinib versus sorafenib in first-line treatment of patients with unresectable hepatocellular carcinoma: a randomised phase 3 non-inferiority trial) and is now widely used not only as first-line but also as second- or third-line chemotherapy [15,16]. The efficacy of cabozantinib as a second-line or third-line treatment was demonstrated by the phase 3 CELESTIAL study (Cabozantinib versus placebo in patients with advanced hepatocellular carcinoma who have received prior sorafenib: results from the randomized phase 3 CELESTIAL trial) [9], and it is now available in some areas and expected to be approved in other countries. Ramucirumab is another second-line systemic therapy shown to be effective in a selected subgroup of patients with alpha-fetoprotein (AFP) >400 ng/mL [10]. Therefore, patients with advanced-stage HCC have the option of receiving two or more systemic therapies in a sequential manner in the near future.

Considering this background, the starting criteria for systemic therapy may change to identify and select good candidates who can tolerate two or more systemic therapies sequentially. Accordingly, the purpose of this study was to identify predictors of good candidates for second-line treatment in patients treated with sorafenib. We first focused on predictors to meet the inclusion criteria of the RESORCE study and further extended the criteria to be stricter for liver function, such as Child–Pugh score 5 or ALBI grade 1.

## 2. Methods

### 2.1. Patients

This is a retrospective cohort study of consecutive HCC patients who were treated with sorafenib in our hospital from July 2009 to April 2017. Diagnosis of HCC was made by pathological or imaging diagnosis, such as dynamic computer tomography (CT), gadolinium ethoxybenzyl diethlenetriamine pentaacetic acid (Gd-EOB-DTPA)-enhanced liver magnetic resonance imaging (Gd-EOB MRI), or sonazoid contrast-enhanced ultrasonography, according to the guideline of the American Association for the Study of Liver Diseases (AASLD) [17,18]. HCC was defined as evidence of a classic pattern-dense staining in the arterial phase and washout in the portal vein phase. The inclusion criteria for the study were as follows: Child–Pugh A at the start of sorafenib administration and discontinuation of sorafenib treatment at the time of database lock (June 2017). A total of 190 patients were included in the analysis. The following patients were excluded from the study: 20 patients with Child–Pugh B at the start of sorafenib therapy and 20 patients who were continuously taking sorafenib at the time of database lock. Written, informed consent was obtained from each patient. This study was approved by the institutional ethics committee (ethics committee for clinical studies of Musashino Red Cross Hospital: Rinshoukenkyu-Rrinrishinsa-Iinkai (in Japanese), approval number 28077) in accordance with the Declaration of Helsinki [19,20].

### 2.2. Sorafenib Treatment

The decision to start sorafenib treatment was based on clinical guidelines in Japan [21]. Briefly, the guidelines state that HCC must be unresectable or unable to be ablated because it is locally advanced or in an advanced stage (Barcelona Clinic Liver Cancer stage C) with distant metastasis or vascular infiltration, or was refractory to transarterial chemoembolization (TACE). All patients were classified as Eastern Cooperative Oncology Group performance status 0–1 [18,22]. 

All patients were treated with sorafenib monotherapy without coadministration of any other systemic chemotherapy. In principle, sorafenib was administered at a starting dose of 800 mg/day. However, in elderly patients >80 years old or patients with body weight <50 kg, the sorafenib dosage was reduced to 400 mg at the discretion of the physician. 

### 2.3. Data Collection

At the start of sorafenib therapy, demographics, radiological findings, and laboratory data were obtained. Liver function was evaluated using the Child–Pugh classification and ALBI grade at baseline 4 weeks after the administration of sorafenib and also at the time of sorafenib discontinuation. ALBI grade was determined by the following formula:ALBI score = (log 10 bilirubin (μmol/L) × 0.66) + (albumin (g/L) × (−0.0852))(1)

The grade was classified into the following three levels: Grade 1: ≤−2.60, Grade 2: <−2.60 to ≤−1.39, and Grade 3: >−1.39 [4]. Tumor volume was evaluated by tumor burden, which is the sum of the maximum intrahepatic tumor size and the number of intrahepatic lesions. CT examinations were performed at baseline, one month after sorafenib introduction, and every 2–3 months of follow-up thereafter. The efficacy was evaluated based on the modified response evaluation criteria in solid tumors (modified RECIST) [23]. Medical examinations and blood tests were performed every 1–2 weeks after the introduction of sorafenib. Laboratory data were also obtained at the time of sorafenib discontinuation.

### 2.4. Definition of Criteria of a Good Candidate for Second-Line Treatment

The criteria of a good candidate for second-line treatment were defined as follows. The first definition was the same as the inclusion criteria of the RESORCE study. The second and third definitions were stricter on liver function. The second definition was the inclusion criteria of the RESORCE study plus Child–Pugh score 5, and the third definition was the inclusion criteria of the RESORCE study plus ALBI grade 1.

### 2.5. Statistical Analysis

Fisher’s test was used to test categorical variables. Kolmogorov–Smirnov test was used to test normality, and the unpaired Student’s *t*-test or Mann–Whitney *U* test was used for continuous variables. A *p* value < 0.05 was considered statistically significant. Overall survival was evaluated using the Kaplan–Meier curve, and a comparison between groups was evaluated using the log-rank test as univariate analysis. For the multivariate analysis, a Cox proportional hazard model was used. The best cut-off values in receiver operating characteristic (ROC) analysis were determined by the Youden index. All statistical analyses were performed using Easy R (EZR) version 1.29 (Saitama Medical Center, Jichi Medical University, Saitama, Japan), which is a graphical user interface for R (the R Foundation for Statistical Computing, Vienna, Austria) [24].

## 3. Results

### 3.1. Baseline Data

The patient background data at baseline are shown in Table 1. The average age was 72.0 ± 9.4 years, and 149 (78%) patients were male. The average ALBI score before sorafenib administration was −2.3 ± 0.43, with 50 patients classified as ALBI grade 1 (26%) and 140 patients classified as ALBI grade 2 (74%). For the Child–Pugh score, 102 patients (54%) were classified as Child–Pugh score 5 (A5), and 88 patients (46%) were classified as Child–Pugh score 6 (A6). Major vascular invasion was observed in 45 patients (24%) and distant metastasis in 76 (40%) patients. The median maximum intrahepatic tumor size was 38 mm, and 83 patients (43%) had a tumor of 50 mm or larger. There were 130 patients (68%) with four or more intrahepatic lesions, and the mean tumor burden was 12 ± 6. The median AFP value was 181 ng/mL, and the median prothrombin induced by vitamin K absence II (PIVKA-II) value was 319 mAU/mL.

### 3.2. Prognosis According to Baseline Liver Function

The median overall survival (OS) was 13.9 (range 9.4–16.4) months, and the median progression-free survival (PFS) was 4.0 (3.3–5.4) months. Patients with Child–Pugh A5 at the time of sorafenib introduction had significantly prolonged OS compared with that of patients with A6 (18.6 vs. 7.9 months, *p* = 0.0004) (Figure 1a). Also, A5 patients had significantly prolonged PFS compared with that of A6 patients (4.7 vs. 3.7 months, *p* = 0.03) (Figure 1b).

Among Child–Pugh A5 patients, 49% were ALBI grade 1 and 51% were grade 2. All Child–Pugh A6 patients were grade 2. Patients with ALBI grade 1 at the time of sorafenib introduction had significantly prolonged OS compared with grade 2 (21.6 vs. 9.3 months, *p* = 0.001) (Figure 1c) and also had significantly prolonged PFS (6.4 vs. 3.6 months, *p* = 0.03) (Figure 1d). Variables by Child-Pugh score and ALBI grade are shown in Appendix A.

### 3.3. Frequency of Patients Fulfilling Criteria as a Good Candidate for Second-Line Treatment 

The reason for sorafenib discontinuation was due to radiological PD in 80 cases (42%). At the time of sorafenib discontinuation, liver function was within Child–Pugh A in 143 patients (75%), and predefined tolerability criteria of the RESORCE study was fulfilled in 113 cases (59%). Among the total of 190 patients, 55 (29%) met all three criteria, 46 (24%) met only two, and 79 (42%) met only one. There were 10 patients (5%) who did not satisfy any criteria. In addition to the inclusion criteria of the RESORCE study, 24 patients (13%) were within Child–Pugh A5, and 11 patients (5.8%) were within ALBI grade 1 (Figure 2).

### 3.4. Predictors for Fulfilling the RESORCE Study Inclusion Criteria at the Time of Sorafenib Discontinuation

Factors at the time of sorafenib administration were analyzed for association with fulfilling the RESORCE criteria at the time sorafenib was discontinued (Table 2). Using univariate analysis, age, gender, tumor burden, macroscopic vascular invasion (MVI), or presence of distant metastasis were not significant factors. However, albumin (Alb) (odds ratio (OR) 2.3, *p* = 0.05) and ALBI score (OR 0.32, *p* = 0.005) were significant factors. There were no differences in other blood test results, including total bilirubin, prothrombin time (PT) activity, ammonia value, and tumor markers. Laboratory data were also assessed four weeks after sorafenib administration. Changes in albumin (ΔAlb at week 4) and ALBI score from baseline (ΔALBI score at week 4) were significantly associated with fulfilling the RESORCE criteria (OR 3.8, *p* < 0.001 for ΔAlb; OR 0.29, *p* = 0.015 for ΔALBI). 

The best cut-off values for baseline albumin, ALBI score, ΔAlb at week 4, and ΔALBI at week 4 were determined using ROC curve analysis. The best cut-off value was 3.7 g/dL for baseline albumin, −2.33 for baseline ALBI, 0.2 g/dL for ΔAlb at week 4, and 0.255 for ΔALBI at week 4. Factors associated with the higher ΔALBI at week 4 were macroscopic vascular invasion (42% vs. 11%, *p* < 0.001), AFP >300ng/mL (58% vs. 38%, *p* < 0.001), and AFP L3 index >40% (45% vs. 26%, *p* = 0.02). In this study, 37 patients were assessed as progressive disease at week 4. Among them, 21 patients had Δalbumin <−0.2 g/dL at week 4 or ΔALBI <−0.255 at week 4.

Because ALBI score integrates albumin in its calculation, multivariate analysis was performed independently using either of these two factors. The first multivariate model incorporated baseline ALBI and ΔALBI at week 4. Baseline ALBI score <−2.33 (OR 2.5, *p* = 0.01) and ΔALBI score at week 4 <0.255 (OR 4.9, *p* < 0.001) were independent factors (Table 2, multivariate model 1). The second multivariate model incorporated baseline Alb and ΔAlb at week 4. In this model, baseline Alb <3.7 g/dL (OR 3.1, *p* < 0.001) and ΔAlb at week 4 <0.2 g/dL (OR 2.6, *p* = 0.006) were significant (Table 2, multivariate model 2). The association between the baseline albumin and Δalbumin was as follows: in patients with albumin >3.7 g/dL at baseline, 45% had Δalbumin >−0.2 g/dL at week 4, while in patients with albumin <3.7 g/dL at baseline, 37% had Δalbumin >−0.2 g/dL at week 4. The association between the baseline ALBI and ΔALBI was as follows: in patients with ALBI <−2.33 at baseline, 35% had ΔALBI >0.255 at week 4, while in patients with ALBI >−2.33 at baseline, 32% had ΔALBI >0.255 at week 4. These results suggest that ΔALBI or Δalbumin is independent of baseline value of ALBI or albumin.

### 3.5. Predictors for Fulfilling the RESORCE Study’s Inclusion Criteria plus Child–Pugh Score 5 at the Time of Sorafenib Discontinuation

Next, factors at the time of sorafenib administration and at four weeks after sorafenib administration were analyzed for association with fulfilling the RESORCE criteria plus Child–Pugh A5 at the time of sorafenib discontinuation (Table 3). Again, Alb at baseline, ALBI score at baseline, ΔAlb at week 4, and ΔALBI at week 4 were significant in univariate analysis.

The best cut-off value was 3.7 g/dL for baseline albumin, −2.33 for baseline ALBI, 0.1 g/dL for ΔAlb at week 4, and 0.19 for ΔALBI at week 4. Compared to predictors for the RESORCE criteria, the best cut-off value of Alb and ALBI at baseline was not different, but those of ΔAlb at week 4 and ΔALBI at week 4 was smaller to fulfill the RESORCE criteria plus Child–Pugh score 5. 

In a multivariate model incorporating baseline ALBI and ΔALBI at week 4, baseline ALBI score <−2.33 (OR 5.3, *p* = 0.002) and ΔALBI score at week 4 <0.19 (OR 5.9, *p* = 0.002) were independent factors (Table 3, multivariate model 1). The second multivariate model incorporating baseline Alb and ΔAlb at week 4, baseline Alb <3.7 g/dL (OR 4.4, *p* = 0.003) and ΔAlb at week 4 <0.2 g/dL (OR 4.5, *p* = 0.002) were significant factors (Table 3, multivariate model 2). 

### 3.6. Predictors for Fulfilling the RESORCE Study Inclusion Criteria Plus ALBI Grade 1 at the Time of Sorafenib Discontinuation

Finally, factors at the time of sorafenib administration and at four weeks after sorafenib administration were analyzed for any association with fulfilling the RESORCE criteria plus ALBI grade 1 at the time of sorafenib discontinuation (Table 4). In addition to Alb at baseline, ALBI score at baseline, ΔAlb at week 4, and ΔALBI at week 4, tumor burden at baseline (OR 0.87, *p* = 0.01) was also a significant factor. 

The best cut-off value was 3.8 g/dL for baseline albumin, −2.55 for baseline ALBI, 0.1 g/dL for ΔAlb at week 4, and 0.085 for ΔALBI at week 4. Compared to predictors for the RESORCE criteria plus Child–Pugh score 5, the best cut-off value of Alb and ALBI at baseline and ΔALBI at week 4 became stricter to fulfill the RESORCE criteria plus ALBI grade 1. The best cut-off value for tumor burden was 11 or less.

In the multivariate model incorporating baseline tumor burden, ALBI and ΔALBI at week 4, tumor burden of 11 or less (OR 13, *p* = 0.02), baseline ALBI score <−2.55 (OR 13, *p* = 0.003), and ΔALBI score at week 4 <0.085 (OR 20, *p* = 0.008) were independent factors (Table 4, multivariate model 1). The second multivariate model incorporating baseline tumor burden, Alb and ΔAlb at week 4, tumor burden of 11 or less (OR 22, *p* = 0.004), baseline Alb <3.8 g/dL (OR 12, *p* = 0.02), and ΔAlb at week 4 <0.1 g/dL (OR 4.8, *p* = 0.04) were significant (Table 4, multivariate model 2).

Table 5 summarizes the different cut-off value of factors to fulfill various conditions at the time of radiological progressive disease. To fulfill Child–Pugh A5 or ALBI grade 1 at disease progression, liver function at baseline and degree of change in liver function need to be stricter compared to the criteria to fulfill Child–Pugh A.

### 3.7. Predictive Value of These Criteria

The sensitivity of these criteria to satisfy the RESORCE criteria (either or both predictors positive), RESORCE plus Child–Pugh A5 (either or both predictors positive), and RESORCE plus ALBI grade 1 (two or all predictors positive) was 85–93%, 88–100%, and 91%, respectively. The negative predictive value of these predictors was 86–92%, 96–100%, and 99%, respectively. Taken together, these results suggest that sensitivity and negative predictive value of these predictors are high to define patients who are likely to fulfill these three criteria at disease progression (Appendix A). 

## 4. Discussion

Liver function is one of the most important predictors of survival after systemic chemotherapy of unresectable HCC [3,14,25]. The present study confirmed that, among patients treated with sorafenib, those classified as Child–Pugh class A5 had better survival rates compared to those classified as Child–Pugh class A6, and those in ALBI grade 1 had better survival than those in grade 2. Because conserved liver function within Child–Pugh A is the key inclusion criteria in all clinical trials of systemic therapy for HCC, including regorafenib, cabozantinib, and ramucirumab [9,10], the identification of factors which can predict preserved liver function with disease progression of first-line treatment may help identify those patients who would benefit most from treatment with multiple TKIs in a sequential manner. In the present study, 29% of patients met the inclusion criteria for the RESORCE study. Factors in meeting this criteria were a combination of baseline albumin of >3.7 g/dL plus a <0.2 g/dL decrease in albumin at week 4. Alternatively, when the ALBI predictor was used, these factors were a combination of baseline ALBI score of <−2.33 and degree of increase of ALBI score at week 4 of <0.255 at week 4. The albumin or ALBI score at the time of sorafenib administration and Δalbumin or ΔALBI score at week 4 was independently important. In this study, 37 patients were assessed as progressive disease at week 4. Among them, 21 patients had Δalbumin <−0.2 g/dL at week 4 or ΔALBI <−0.255 at week 4. Because this was a retrospective study and regorafenib was not available at that time, some of these patients continued to take sorafenib beyond progressive disease. However, these patients may be candidates to be switched to regorafenib before they progress to Child–Pugh B. We further extended our analysis to define stricter criteria of preserved liver function, i.e., inclusion criteria of the RESORCE study plus Child–Pugh A5 and inclusion criteria of the RESORCE study plus ALBI grade 1. To meet the inclusion criteria of the RESORCE study plus Child–Pugh A5, the value of baseline albumin or ALBI score was identical to the criteria mentioned above but with a stricter degree of change in albumin and ALBI score. To meet the inclusion criteria of the RESORCE study plus ALBI grade 1, the value of baseline albumin or ALBI score became stricter, as did the degree of change in albumin and ALBI score. Furthermore, tumor burden was found to be an additional predictor. In the era where multiple TKI options are available, our results could inform clinicians of the optimal timing to initiate first-line TKI and determine how to deploy multiple TKIs in a sequential manner. 

The strategy to treat patients who do not meet these criteria is the most difficult question. For patients who do not meet the RESORCE criteria due to intolerance to sorafenib, other agents such as lenvatinib or ramucirmab could be the choice for second-line treatment. However, if patients do not meet the criteria due to albumin or ALBI at baseline, the choice of treatment may be limited. The most important point is that we should decide to introduce TKI therapy before the patients fall outside these criteria. In patients who have Δalbumin >−0.2 g/dL or ΔALBI score >0.255 at week 4, radiological assessment for progressive disease should be done more frequently to avoid missing the opportunity to switch to regorafenib.

Multiple TKIs, such as regorafenib, cabozantinib, and ramucirumab, are now available in some countries as a second-line treatment at the time of progression after sorafenib [26]. Lenvatinib is another first-line TKI but now used as a second- or third-line treatment in Japan [15]. Sequential use of these multiple TKIs can prolong the survival of unresectable HCC, although the evidence is yet to be established. Regorafenib was the first TKI that proved to be effective as a second-line treatment. The eligibility criteria for regorafenib administration in the RESORCE study were tolerance to sorafenib and Child–Pugh A. There are several reports with regard to baseline factors at the start of sorafenib to meet this criteria at the time of sorafenib discontinuation. In one study, 36% of patients met these criteria [20], and significant predictors were Child–Pugh A5 and an AST <40 IU/L at baseline [27]. In another report that included 160 patients treated with sorafenib, 30.6% of patients met the criteria, and the absence of MVI and albumin >3.5 g/dL at baseline were predictors [28]. According to a report by Ogasawara et al., 37% of cases met the criteria, and negative predictors were PS1 and Child–Pugh A6 at baseline [29]. In another study utilizing ALBI score, ALBI score <−2.53 was the best predictor [30]. Our results are similar to these previous studies in that baseline liver function, as reflected by albumin or ALBI score, are related in fulfilling the RESORCE criteria. However, the unique aspect of our study is that we examined early changes in liver function at week 4 of sorafenib treatment in addition to baseline factors. To our knowledge, this is the first report involving early changes in liver function to predict if patients would meet appropriate criteria for second-line TKI. Moreover, we defined stricter criterion at disease progression (Child–Pugh A5 or Alb grade 1) because highly preserved liver function at disease progression of first-line treatment may increase the chances of benefiting from second- and third-line TKI in a sequential manner.

The major limitation of this study is that this was a retrospective study, and information about the follow-up therapies is not available. 

Systemic chemotherapy may be targeted to TACE refractory HCC in Barcelona Clinic Liver Cancer (BCLC) stage B and for advanced-stage HCC in BCLC stage C with major vascular invasion or distant metastasis. The proportion of patients who met the RESORCE criteria was 29%, RESORCE criteria plus Child–Pugh A5 was 13%, and RESORCE criteria plus ALBI grade 1 was 5.8%. Notably, predictors in the strictest criteria, i.e., RESORCE criteria plus ALBI grade 1, was baseline ALBI score <−2.55, ΔALBI at week 4 <0.085, and included tumor burden <11. These conditions fall into intermediate-stage BCLC B1 or B2, which are conventionally treated by TACE. It is possible that BCLC stage B patients fulfilling these stringent predictors could do well and survive if they undergo sequential treatment using multiple TKIs. However, future studies are necessary to test this hypothesis. 

## 5. Conclusions

In conclusion, 29% of patients treated with sorafenib met the inclusion criteria for the RESORCE study. Predictive factors of the RESORCE study inclusion criteria were preserved liver function at baseline and less deterioration of liver function during early sorafenib therapy. To fulfill better liver function at disease progression, such as Child–Pugh A5 or ALBI grade 1, liver function at baseline and degree of change in liver function need to be stricter. This information may help optimize treatment strategy using sorafenib and regorafenib in a sequential manner and fully utilize multiple TKIs in the future.

## Figures and Tables

**Figure 1 cancers-11-01256-f001:**
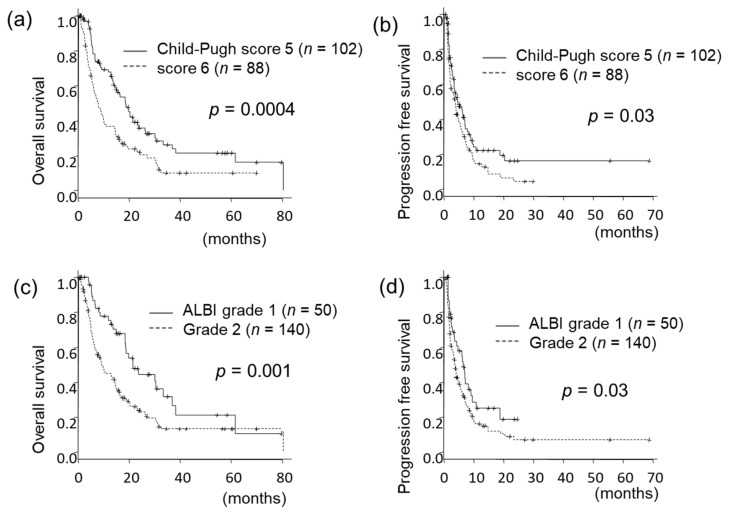
Overall survival (OS) and progression-free survival (PFS) stratified by Child–Pugh score or ALBI grade. Patients within Child–Pugh score 5 at baseline had significantly prolonged OS (**a**) and PFS (**b**) compared with those with score 6. Patients with ALBI grade 1 had significantly prolonged OS (**c**) and PFS (**d**) compared with those with grade 2.

**Figure 2 cancers-11-01256-f002:**
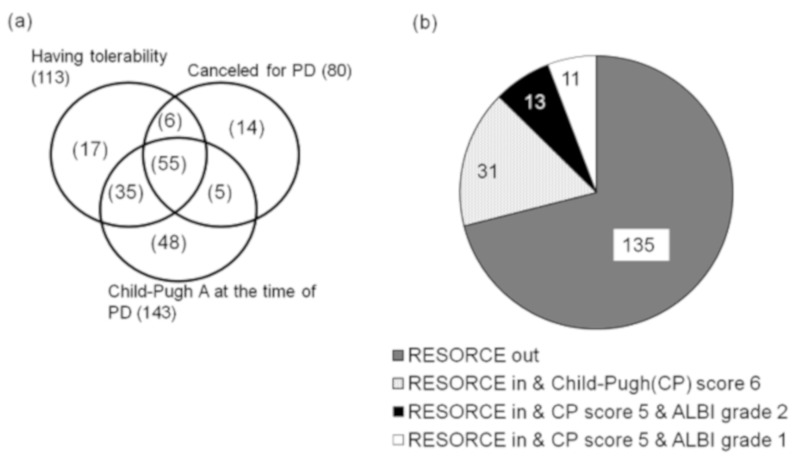
Proportion of patients fulfilling the criteria to be a good candidate for second-line treatment. (**a**) Proportion of patients in each RESORCE criteria. (**b**) Patients within Child–Pugh score 5 and within ALBI grade 1.

**Table 1 cancers-11-01256-t001:** Background of patients.

Variables	*n* = 190
Age: years, mean ± SD	72 ± 9
Male: *n* (%)	149 (78%)
Maximum size of HCC nodule in the liver: mm, median (range in IQR)	38 (20–74)
Number of HCC nodules: 1/2–3/4 or more, *n* (%)	28 (15%)/32 (17%)/130 (68%)
Tumor burden (maximum size + number of HCC nodules): mean ± SD	12 ± 6
BCLC stage: B/C, *n* (%)	86 (45%)/104 (55%)
Presence of MVI: *n* (%)	45 (24%)
Presence of metastasis: *n* (%)	76 (40%)
Albumin: g/dL, mean ± SD	3.6 ± 0.4
Total bilirubin: g/dL, mean ± SD	0.9 ± 0.5
Prothrombin time: %, mean ± SD	98 ± 16
Child–Pugh score: 5/6, *n* (%)	102/88 (54%/46%)
ALBI predictor, mean ± SD	−2.30 ± 0.42
ALBI grade: 1/2, *n* (%)	50/140 (26%/74%)
AFP: ng/mL, median (range in IQR)	181(13–3080)
AFP L3 index: % (range in IQR)	18 (5.8–49)
PIVKA-II: mAU/mL, median (range in IQR)	319 (49–2308)

AFP: alpha-fetoprotein, AFP L3 fraction: Lens culinaris agglutinin-reactive fraction of AFP, ALBI: albumin–bilirubin, BCLC: Barcelona Clinic Liver Cancer stage, HCC: hepatocellular carcinoma, IQR: interquartile range, MVI: major vascular invasion, PIVKA-II: prothrombin induced by vitamin K absence II, SD: standard deviation.

**Table 2 cancers-11-01256-t002:** Factors to fulfill the RESORCE study’s inclusion criteria at the time of radiological progressive disease.

Factors	Univariate		Multivariate Model 1: ALBI				Multivariate Model 2: Alb			
	OR	*p* Value	Cut-Off Value	OR	95% CI	*p* Value	Cut-Off Value	OR	95% CI	*p* Value
Baseline factors										
Age	0.67	0.37								
Male	0.63	0.27								
Tumor burden	1.0	0.91								
MVI	0.63	0.26								
Metastasis	0.91	0.74								
BCLC stage C	0.67	0.19								
Albumin (g/dL)	2.5	0.01	-				>3.7 g/dL	2.7	1.4–5.2	0.004
T.Bil (g/dL)	0.54	0.19								
Prothrombin time (%)	0.99	0.24								
ALBI predictor	3.1	0.005	<−2.33	2.5	1.3–5.2	0.01	-			
AFP (ng/mL)	0.71	0.34								
PIVKA-II (mAU/mL)	1.0	0.54								
Factors at week 4										
ΔAlb at week4 (g/dL)	3.8	<0.001	-				<−0.2 g/dL	2.6	1.3–5.2	0.005
ΔT.Bil at week4 (mg/dL)	0.56	0.35								
ΔALBI at week4	0.29	0.015	<0.255	4.9	2.1–11	0.0001	-			

AFP: alpha-fetoprotein, Alb: albumin, ALBI: albumin–bilirubin, BCLC: Barcelona Clinic Liver Cancer stage, 95% CI: 95% confidence interval, MVI: major vascular invasion, OR: odds ratio, PIVKA-II: prothrombin induced by vitamin K absence II, T.Bil: total bilirubin.

**Table 3 cancers-11-01256-t003:** Factors to fulfill the RESORCE study’s inclusion criteria plus Child–Pugh score 5 at the time of radiological progressive disease.

Factors	Univariate		Multivariate Model 1: ALBI				Multivariate Model 2: Alb			
	OR	*p* Value	Cut-Off Value	OR	95% CI	*p* Value	Cut-Off Value	OR	95% CI	*p* Value
Baseline factors										
Age	1.0	0.20								
Male	2.1	0.26								
Tumor burden	0.96	0.23								
MVI	0.26	0.07								
Metastasis	0.92	0.86								
BCLC stage C	0.55	0.17								
Albumin (g/dL)	2.8	0.04	-				>3.7 g/dL	2.9	1.2–7.2	0.02
T.Bil (g/dL)	0.13	0.01								
Prothrombin time (%)	1.0	0.44								
ALBI predictor	0.21	0.005	<−2.33	5.4	1.9–16	0.002	-			
AFP (ng/mL)	1.0	0.63								
PIVKA-II (mAU/mL)	1.0	0.23								
Factors at week 4										
ΔAlb at week4 (g/dL)	21	<0.001	-				<−0.1 g/dL	4.1	1.6–11	0.003
ΔT.Bil at week4 (mg/dL)	0.92	0.82								
ΔALBI at week4	0.13	0.005	<0.19	5.9	1.9–19	0.003	-			

AFP: alpha-fetoprotein, Alb: albumin, ALBI: albumin–bilirubin, BCLC: Barcelona Clinic Liver Cancer stage, 95% CI: 95% confidence interval, MVI: major vascular invasion, OR: odds ratio, PIVKA-II: prothrombin induced by vitamin K absence II, T.Bil: total bilirubin.

**Table 4 cancers-11-01256-t004:** Factors to fulfill the RESORCE study’s inclusion criteria plus ALBI grade 1 at the time of radiological progressive disease.

Factors	Univariate		Multivariate Model 1: ALBI				Multivariate Model 2: Alb			
	OR	*p* Value	Cut-Off Value	OR	95% CI	*p* Value	Cut-Off Value	OR	95% CI	*p* Value
Baseline factors										
Age	0.99	0.18								
Male	1.3	0.78								
Tumor burden	0.87	0.01	<11	13	1.5–118	0.02	<11	12	1.5–12	0.0002
MVI	0.81	0.77								
metastasis	0.36	0.11								
BCLC stage C	0.43	0.90								
Albumin (g/dL)	9.8	0.002	-				>3.8 g/dL	8.7	2.0–39	0.004
T.Bil (g/dL)	0.42	0.34								
Prothrombin time (%)	1.0	0.53								
ALBI predictor	0.07	0.002	<−2.55	14	2.4–76	0.003	-			
AFP (ng/mL)	1.0	0.79								
PIVKA-II (mAU/mL)	1.0	0.43								
Factors at week 4										
ΔAlb at week4 (g/dL)	21	<0.001	-				<−0.1 g/dL	5.4	1.2–24	0.03
ΔT.Bil at week4 (mg/dL)	0.62	0.27								
ΔALBI at week4	0.08	0.01	<0.085	20	2.2–177	0.008	-			

AFP: alpha-fetoprotein, Alb: albumin, ALBI: albumin–bilirubin, BCLC: Barcelona Clinic Liver Cancer stage, 95% CI: 95% confidence interval, MVI: major vascular invasion, OR: odds ratio, PIVKA-II: prothrombin induced by vitamin K absence II, T.Bil: total bilirubin.

**Table 5 cancers-11-01256-t005:** The cut-off value of factors to fulfill various conditions at the time of radiological progressive disease.

Factors	RESORCE Study’s Criteria	RESORCE Study’s Criteria + Child–Pugh Score5	RESORCE Study’s Criteria +ALBI Grade1
Baseline albumin (g/dL)	3.7	3.7	3.8
Baseline ALBI predictor	−2.33	−2.33	−2.55
ΔAlb at week 4 (g/dL)	−0.2	−0.1	−0.1
ΔALBI at week 4	0.255	0.19	0.085
Tumor burden	NS	NS	11

Alb: albumin, ALBI: albumin–bilirubin.

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
