# Peer review of "Baseline and Early Predictors of Good Patient Candidates for Second-Line after Sorafenib Treatment in Unresectable Hepatocellular Carcinoma"

_cancers, 2019, doi:10.3390/cancers11091256_

Round 1

Reviewer 1 Report

This article shows that baseline and early changes in liver function (albumin or ALBI score) were identified as predictors of good candidates with unresectable HCC for second-line treatment after disease progression during sorafenib treatment. However, there are some problems in this article.

Major points:

1.     In this study (190 patients), the proportion of patients who met the criteria for the RESORCE study was 29% (55 patients), RESORCE criteria plus Child-Pugh A5 was 13% (24 patients), and RESORCE criteria plus ALBI grade 1 was 5.8% (11 patients). As the authors mentioned, albumin or ALBI score at baseline was predictors of good candidates for second-line treatment after disease progression during sorafenib treatment like previous reports. The authors described that the unique aspect in this study is early changes (albumin or ALBI score) at week 4 of sorafenib treatment. How many patients who have predictors to satisfy the RESORCE study inclusion criteria, RESORCE criteria plus Child-Pugh A5, and RESORCE criteria plus ALBI grade 1 are detected in this study?

2.     This article shows that patients withΔalbumin <-0.2 g/dL at week 4 orΔALBI <-0.255 at week 4 can be switched from sorafenib to regorafenib. How many patients who were assessed as progressive disease at week 4 in such patients are detected in this study?

3.     I think albumin or ALBI score at the time of sorafenib administration as important factors, but not Δalbumin or ΔALBI score at week 4. In this study, baseline albumin and Δalbumin are independent predictors of good patient candidates for second-line sorafenib treatment. For example, how many patients who have Δalbumin >-0.2 g/dl at week 4 are detected in patients with albumin >3.7 g/dL at baseline?

4.     How do the authors treat for patients who do not have these criteria according to this results?

Minor points:

1.     In the title, the authors should add the sentences “in unresectable hepatocellular carcinoma”.

2.     In the prognosis according to baseline liver function section, the authors demonstrated that patients with Child-Pugh A5 or ALBI grade 1 at the time of sorafenib induction had significantly prolonged overall survival or progressive free-survival compared with that of the patients with Child-Pugh A6 or ALBI grade 2. Were there differences in the patient characteristics among each groups? The authors should describe the details.

3.     In Figure 2 (b), the color with RESORCE in & CP score 6 is similar that with RESORE in & CP score 5 & ALBI grade 2.

4.     On page 7, line 221, the words (OR 0.87, p = 0.01) should be moved to the position after “tumor burden at baseline”.

5.     In Table 4, the cut off value in ΔALBL at week 4 should be change from <0.09 to <0.085.

6.    There is no explain in regard to Table 5 in the text.

Author Response

Author's Notes

This article shows that baseline and early changes in liver function (albumin or ALBI score) were identified as predictors of good candidates with unresectable HCC for second-line treatment after disease progression during sorafenib treatment. However, there are some problems in this article.

Thank you very much for your constructive suggestions and comments. We made additional analysis according to your suggestion.

Major points:

In this study (190 patients), the proportion of patients who met the criteria for the RESORCE study was 29% (55 patients), RESORCE criteria plus Child-Pugh A5 was 13% (24 patients), and RESORCE criteria plus ALBI grade 1 was 5.8% (11 patients). As the authors mentioned, albumin or ALBI score at baseline was predictors of good candidates for second-line treatment after disease progression during sorafenib treatment like previous reports. The authors described that the unique aspect in this study is early changes (albumin or ALBI score) at week 4 of sorafenib treatment. How many patients who have predictors to satisfy the RESORCE study inclusion criteria, RESORCE criteria plus Child-Pugh A5, and RESORCE criteria plus ALBI grade 1 are detected in this study?

To satisfy the RESORCE study inclusion criteria, the number of patients fulfilled both predictors of baseline and Δalbumin, one predictor, and no predictor was 30, 102, and 58, respectively. The sensitivity to satisfy the RESORCE criteria was 85% if patients had one or both predictors. On the other hand, 86% of patients who fulfilled no predictor was out of RESORCE criteria at disease progression. For baseline and ΔALBI, the number of patients who fulfilled both predictors, one predictor, and no predictor by was 48, 89, and 53, respectively. The sensitivity was 93% if patients had one or both predictors. On the other hand, 92% of patients who fulfilled no predictor was out of RESORCE criteria at disease progression.

To satisfy the RESORCE plus Child-Pugh A5, the number of patients fulfilled both predictors of baseline and Δalbumin, one predictor, and no predictor was 23, 94, and 73, respectively. The sensitivity was 88% if patients had one or both predictors. The 96% of patients who fulfilled no predictor was out of RESORCE plus Child-Pugh A5 criteria. For baseline and ΔALBI, the number of patients who fulfilled both predictors, one predictor, and no predictor by was 43, 88, and 59, respectively. The sensitivity was 100% if patients had one or both predictors. All patients (100%) who fulfilled no predictor was out of RESORCE plus Child-Pugh A5 criteria at disease progression.

To satisfy the RESORCE plus ALBI grade 1, the number of patients fulfilled all three predictors of baseline and Δalbumin, and tumor burden, two predictors, one predictor, and no predictor was 11, 46, 77, and 56, respectively. The sensitivity was 91% if patients had two predictors. The 99% of patients who fulfilled one or no predictor was out of RESORCE plus ALBI grade 1 at disease progression. For baseline and ΔALBI, and tumor burden, the number of patients who fulfilled all three predictors, two predictors, one predictor, and no predictor was 13, 40, 79, and 58, respectively. The sensitivity was 91% if patients had two predictors. The 99% of patients who fulfilled one or no predictor was out of RESORCE plus ALBI grade 1 at disease progression.

Taken together these results suggest that sensitivity and negative predictive value of these predictors were high to define patients who are likely to fulfill these three criteria at disease progression.

These results may be incorporated in the result section. However, we are afraid that the text becomes too long. Therefore, we would like to summarize these results in short as follows.

‘The sensitivity of these criteria to satisfy the RESORCE criteria (either or both predictors positive), RESORCE plus Child-Pugh A5 (either or both predictors positive), and RESORCE plus ALBI grade 1 (two or all predictors positive) was 85-93%, 88-100%, and 91%, respectively. The negative predictive value of these predictors was 86-92%, 96-100%, and 99%, respectively. Taken together these results suggest that sensitivity and negative predictive value of these predictors were high to define patients who are likely to fulfill these three criteria at disease progression.’ (page10, lines 256-262). This data and comment are now presented in the result section.

This article shows that patients with Δalbumin <-0.2 g/dL at week 4 or ΔALBI <-0.255 at week 4 can be switched from sorafenib to regorafenib. How many patients who were assessed as progressive disease at week 4 in such patients are detected in this study?

In this study, 37 patients were assessed as progressive disease at week 4. Among them, 21 patients had Δalbumin <-0.2 g/dL at week 4 or ΔALBI <-0.255 at week 4. Because this was a retrospective study, and regorafenib was not available at that time, some of these patients continued to take sorafenib beyond progressive disease. However, these patients may be a candidate to switched to regorafenib before they progress to Child-Pugh B (page10, lines 278-283). This data and comment are now presented in the result and discussion section.

I think albumin or ALBI score at the time of sorafenib administration as important factors, but not Δalbumin or ΔALBI score at week 4. In this study, baseline albumin and Δalbumin are independent predictors of good patient candidates for second-line sorafenib treatment. For example, how many patients who have Δalbumin >-0.2 g/dl at week 4 are detected in patients with albumin >3.7 g/dL at baseline?

The association between the baseline albumin and Δalbumin was as follows: in patients with albumin >3.7 g/dL at baseline, 45% had Δalbumin >-0.2 g/dl at week 4 while in patients with albumin <3.7 g/dL at baseline, 37% had Δalbumin >-0.2 g/dl at week 4. The association between the baseline ALBI and ΔALBI is as follows: in patients with ALBI< -2.33 at baseline, 35% had ΔALBI> 0.255 at week 4 while in patients with ALBI> -2.33 at baseline, 32% had ΔALBI> 0.255 at week 4 (page7, lines 203-207). This data and comment are now presented in the result section.

These results suggest that ΔALBI or Δalbumin was independent of baseline value of ALBI or albumin (page7, lines 207). Therefore, albumin or ALBI score at the time of sorafenib administration and Δalbumin or ΔALBI score at week 4 was independently important (page10, lines 277-278). This data and comment are now presented in the result and discussion section.

How do the authors treat for patients who do not have these criteria according to this results?

The strategy to treat patients who do not have these criteria is the most difficult question. For patients who did not meet RESORCE criteria due to intolerance to sorafenib, other agents such as lenvatinib or ramucirmab could be the choice of the second-line. However, if the patients did not meet the criteria due to albumin or ALBI at baseline, the choice of treatment may be limited. The most important point is that we should decide to introduce TKI therapy before the patients fall outside these criteria. In patients who have Δalbumin >-0.2 g/dl or ΔALBI score > 0.255 at week 4, radiological assessment for progressive disease should be done more frequently to avoid missing the opportunity to switch to regorafenib. These comments are now presented in the discussion section (page11, lines 293-300).

Minor points:

In the title, the authors should add the sentences “in unresectable hepatocellular carcinoma”.

Thank you for the suggestion. We now added “in unresectable hepatocellular carcinoma” in the title.

In the prognosis according to baseline liver function section, the authors demonstrated that patients with Child-Pugh A5 or ALBI grade 1 at the time of sorafenib induction had significantly prolonged overall survival or progressive free-survival compared with that of the patients with Child-Pugh A6 or ALBI grade 2. Were there differences in the patient characteristics among each groups? The authors should describe the details.

The patient characteristics between Child-Pugh A5 and A6, or ALBI grade 1 and 2 were compared, and provided as a supplementary table 1 and 2. For comparison between Child-Pugh A5 and A6, albumin, bilirubin, prothrombin time, and ALBI was different as expected. In addition, AFP was higher in patients with Child-Pugh A6. For comparison between ALBI grade 1 and 2, albumin, bilirubin, prothrombin time, and ALBI was different as expected. In addition, tumor burden and AFP were higher in patients with ALBI grade 2.

In Figure 2 (b), the color with RESORCE in & CP score 6 is similar that with RESORE in & CP score 5 & ALBI grade 2.

In Figure 2 (b), the color with RESORCE in & CP score 6 was changed to differentiate from RESORE in & CP score 5 & ALBI grade 2.

On page 7, line 221, the words (OR 0.87, p = 0.01) should be moved to the position after “tumor burden at baseline”.

The words (OR 0.87, p = 0.01) was moved to the position after “tumor burden at baseline” as suggested (page8, lines 233).

In Table 4, the cut off value in ΔALBL at week 4 should be change from <0.09 to <0.085.

In Table 4, the cut off value in ΔALBI at week 4 was changed from <0.09 to <0.085 as suggested.

There is no explain in regard to Table 5 in the text.

The explanation of Table 5 was added at the last part of the results section. (page10, lines 252-255)

Reviewer 2 Report

Comments to the authors

This study retrospectively investigated the predictors of good candidates for second-line treatment after disease progression during sorafenib treatment in patients with unresectable hepatocellular carcinoma (HCC).

This is a well-written and interesting paper that addresses an important clinical issue. However, it includes some crucial problems that need to be clarified.

1.    What factors were associated the ΔALBI at week 4? Please clarify.

2.    In statistical analysis section, please describe how to determine the best cut-off values in ROC analysis. Threshold value closest to upper left corner of ROC curve or Youden index?

3.    I recommend the authors to cite the following paper in the introduction section:” Tada T, et al.  J Gastroenterol Hepatol. 2019 Jun;34(6):1066-1073.

Author Response

Thank you very much for the suggestions of important points.

Factors associated with the higher ΔALBI at week 4 were macroscopic vascular invasion (42 vs. 11%, p<0.001), AFP >300ng/ml (58 vs. 38 %, p<0.001), and AFP L3 index (45 vs. 26%, p=0.02). This data is now presented in the result section. (page6, lines 192-196)

The best cut-off values in ROC analysis was determined by Youden index. This is now presented in the method section. (page3, lines 130-131)

The following paper was added at the introduction section as suggested:”Tada T, et al. J Gastroenterol Hepatol. 2019 Jun;34(6):1066-1073.”

Reviewer 3 Report

In their manuscript „Baseline and early predictors of good patient candidates for second-line Sorafenib treatment” Takada et al. report their retrospective analysis of 190 patients treated with Sorafenib for Hepatocellular carcinoma (HCC). The aim of this study was to identify predictors for good candidates for second line treatment after disease progression on Sorafenib. These were defined by meeting the inclusion criteria of the RESORCE study, which evaluated Regorafenib in a second line setting, alone or in combination with very good liver function (as defined by Child Pugh A5 or ALBI grade 1). They identified preserved liver function at baseline before the start of Sorafenib therapy and change of liver function after four weeks as most relevant factors for fulfilling the criteria.

The manuscript is well written and clearly presents the results. The study is of relevance for researchers and clinicians in the field. Although nearby, the main finding that excellent liver function at baseline indicates good suitability for later line treatments. However, the study also has some limitations, mostly its retrospective nature. Also, no information if the patients received follow-up therapies are available. More detailed comments are provided in the comments to the authors.

Author Response

Thank you very much for your important comments.

  We recognize that the retrospective nature of the study and the absence of information about the follow-up therapies are the major limitation of this study. This is now acknowledged in the discussion section. (page11, lines 322-323)

Reviewer 4 Report

   Authors have assessed the identify predictors for second-line treatment after disease progression during sorafenib treatment. This work showed that the inclusion criteria of RESORCE study plus Child-Pugh A5 and inclusion criteria of RESORCE study plus ALBI grade 1 were useful for second-line treatment. Especially, this work showed that inclusion criteria of RESORCE study plus ALBI grade 1 became stricter. These results make easy to predict disease progression and these will help optimize treatment strategy by using sorafenib and regorafenib in sequential manner. This work is very interesting and important for the strategy of HCC.

Author Response

Thank you very much for your encouraging comments.

Round 2

Reviewer 1 Report

Revised manuscript was well-addressed for reviewers' comments and well-written. However, some corrections are required in the revised manuscript.

On page 6, line195, the authors described AFP >300 ng/mL (58 vs. 38 %, p<0.001), and AFP L3 index (45 vs. 26%, p=0.02). How did the authors determine the cut-off value of 300 ng/mL? The median AFP value was 181 ng/mL in this study. The authors should describe this reason. In addition, there were no data of AFPL3 in Table 1. If the authors describe AFP L3 index (45 vs. 26%, p=0.02), AFPL3 data should be shown in Table 1. In Table 5, the title was “The cut-off of baseline factors to fulfill various conditions ~”. However, Δ Alb at week 4 and Δ ALBI at week 4 were not baseline factors. On page 10, line 253, the sentence “Table 5 summarized the different cut-off value of baseline factors~” should also be corrected.

Author Response

On page 6, line195, the authors described AFP >300 ng/mL (58 vs. 38 %, p<0.001), and AFP L3 index (45 vs. 26%, p=0.02). How did the authors determine the cut-off value of 300 ng/mL? The median AFP value was 181 ng/mL in this study. The authors should describe this reason. In addition, there were no data of AFPL3 in Table 1. If the authors describe AFP L3 index (45 vs. 26%, p=0.02), AFPL3 data should be shown in Table 1.

Thank you for the additional comments as above. The cut-off value of AFP and AFP L3 were determined by ROC analysis with Youden index as 294 and 37.3, respectively, and the approximate value of 300 and 40, respectively. This is presented in the method and result section (page3, lines 130-131) (page6, lines 192-196). The data of AFP L3 is now added in Table 1.

In Table 5, the title was “The cut-off of baseline factors to fulfill various conditions ~”. However, Δ Alb at week 4 and Δ ALBI at week 4 were not baseline factors. On page 10, line 253, the sentence “Table 5 summarized the different cut-off value of baseline factors~” should also be corrected

Thank you for the comment. We have deleted the word ‘baseline’. The title of table 5 was revised as ‘The cut-off value of factors to fulfill various conditions at the time of radiological progressive disease.’ The sentence “Table 5 summarized the different cut-off value of baseline factors~” was revised as “Table 5 summarized the different cut-off value of factors~ (page10, lines 252-255).”

Round 3

Reviewer 1 Report

2-nd revised manuscript was well-addressed for reviewers' comments and well-written. If possible, only minor revision described below is needed.

On page, 10, lines 257-262 (3.7. Predictive value of these criteria), predictive value of these criteria is a very complex section. Therefore, I would like a supplementary table to lead the readers to understand this section.

Author Response

2-nd revised manuscript was well-addressed for reviewers' comments and well-written. If possible, only minor revision described below is needed.

On page, 10, lines 257-262 (3.7. Predictive value of these criteria), predictive value of these criteria is a very complex section. Therefore, I would like a supplementary table to lead the readers to understand this section.

Response: Thank you for the comment. Predictive value of these criteria were summarized, and provided as a supplementary table 3, 4 and 5. The sensitivity of these criteria to satisfy the RESORCE criteria (either or both predictors positive), RESORCE plus Child-Pugh A5 (either or both predictors positive), and RESORCE plus ALBI grade 1 (two or all predictors positive) was 85-93%, 88-100%, and 91%, respectively. The negative predictive value of these predictors was 86-92%, 96-100%, and 99%, respectively.